# Presence of Choroidal Caverns in Patients with Posterior and Panuveitis

**DOI:** 10.3390/biomedicines11051268

**Published:** 2023-04-25

**Authors:** Tedi Begaj, Amy Yuan, Ines Lains, Ashley Li, Samuel Han, Gayatri Susarla, Ravi Parikh, Lucia Sobrin

**Affiliations:** 1Department of Ophthalmology, Massachusetts Eye and Ear, Harvard Medical School, Boston, MA 02114, USA; 2Retina Department, University of Washington, Seattle, WV 98195, USA; 3Manhattan Retina and Eye Consultants, New York, NY 10075, USA; 4Department of Ophthalmology, New York University School of Medicine, New York, NY 10016, USA

**Keywords:** choroidal caverns, indocyanine green angiography, optical coherence tomography, panuveitis, posterior uveitis, uveitis

## Abstract

Choroidal caverns (CCs) have been described in association with age-related macular degeneration and pachychoroid disease. However, it is unknown if caverns are found in patients with chronic non-infectious uveitis (NIU). Herein, we evaluated patients with NIU who had optical coherence tomography and indocyanine green angiography for CCs. Clinical and demographic characteristics were extracted from the chart review. Univariate and multivariate mixed-effects logistical models were used to assess the association between clinical and demographic factors and the presence of CCs. One hundred thirty-five patients (251 eyes) met the inclusion criteria: 1 eye had anterior uveitis, 5 had intermediate uveitis, 194 had posterior uveitis, and 51 had panuveitis. The prevalence of CCs was 10%. CCs were only observed in patients with posterior and panuveitis, with a prevalence of 10.8% and 7.8%, respectively. Multifocal choroiditis (MFC) was the type of uveitis where CCs were most frequently observed, with 40% of eyes with MFC having CCs. In addition, male sex (*p* = 0.024) was associated with CCs. There was no significant difference in the degree of intraocular inflammation or mean subfoveal choroidal thickness between CC+ and CC− eyes. This is the first study to describe CCs in uveitis. Overall, these findings suggest that caverns may be a sequela of structural and/or vascular perturbations in the choroid from uveitis.

## 1. Introduction

Uveitis captures a broad spectrum of disease entities caused by infectious or immune-mediated etiologies. The Standardization of Uveitis Nomenclature (SUN) international group has established a framework for the classification of uveitis entities based on the location of inflammation–anterior, intermediate, posterior, and panuveitis [1]. The primary site of inflammation for anterior uveitis is the anterior chamber, affecting the iris and anterior ciliary body, while inflammation of the vitreous is intermediate uveitis. Posterior uveitis refers to retinal or choroidal inflammation, while panuveitis specifies inflammation in all three aforementioned sites.

The choroid can be affected in different ways in posterior and panuveitis. For example, in birdshot chorioretinopathy, hypocyanescent spots on indocyanine green angiography (ICGA) are hypothesized to correspond to choroidal infiltrates initially and can evolve to choroidal atrophy as the disease progresses [2]. Ischemia of the choroid can also be seen in some uveitides [3]. For example, systemic lupus erythematosus can manifest as an occlusive choroidal vasculitis, which can, in turn, lead to vascular attenuation as well as chorioretinal atrophy [4]. Other “white dot syndromes” [5,6,7] have various characteristic choroidal changes, including thickening, hypoperfusion, and progressive thinning.

Pathogenic changes to the choroid have been implicated in a wide variety of chorioretinal conditions. “Choroidal caverns” (CCs) have been identified by spectral-domain optical coherence tomography (OCT) as a novel finding in eyes with geographic atrophy due to age-related macular degeneration (AMD) [8,9]. The authors posit that CCs may be related to the process of choroidal sclerosis and atrophy in AMD. More recent work has found CCs in eyes with pachychoroid disease [10], Best vitelliform macular dystrophy [11], and Stargardt Disease [12]. There has also been the further characterization of CCs into a spectrum of morphological findings based on multimodal imaging [13].

Given the choroidal involvement in various uveitic processes, we hypothesized that choroidal caverns would be found on OCT imaging of these patients. In the current study, we investigated the presence of choroidal caverns in eyes with anterior, intermediate, posterior, and panuveitis.

## 2. Materials and Methods

The study was approved by the Mass General Brigham Institutional Review Board, followed the tenets of the Declaration of Helsinki, and complied with the Health Insurance Portability and Accountability Act. A retrospective review was performed of all patients from 2010 to 2019 with a diagnosis of non-infectious uveitis and concurrent ICGA and OCT enhanced depth imaging (EDI) [73 sections, 30-degree, high scan resolution]. Because different uveitides can have different effects on the choroid, ICGA is critical to confirm that the choroidal caverns seen on OCT are indeed areas devoid of significant blood flow. Furthermore, ICGA was necessary to rule out old regressed choroidal neovascularization. Exclusion criteria included infectious causes of uveitis, as well as presence of concurrent non-inflammatory retinal or choroidal disease; patients with age-related macular degeneration and pachychoroid diseases were specifically excluded. Further exclusion criteria also included poor image quality (approximately 30 patients had poor imaging quality while 59 patients were excluded due to concurrent or non-uveitic retinal disease such as proliferative diabetic retinopathy, inherited retinal degeneration, etc.). Imaging was evaluated at the initial consultation.

Charts of the included participants were reviewed, and the following information was collected: age, sex, ethnicity, underlying diagnosis, duration of disease, degree of inflammation at the time of imaging, and treatment. The degree of anterior chamber [1] and vitreous cells [14] were ascertained by clinical examination per established criteria and recorded in the medical record by one examiner (L.S). Treatment administered at the time of imaging was recorded as either systemic, local, or none. Systemic treatment included oral, subcutaneous, and intravenous corticosteroids or steroid-sparing immunomodulatory therapy. Local therapy included topical, periocular, or intravitreal corticosteroids.

For all included participants, ICGA images were acquired using either the Optos California (Optos, Dunfermline, Scotland), Topcon TRC-50IX fundus camera (Topcon USA, Paramus, NJ, USA), or Spectralis instrument (Heidelberg Engineering Inc., Franklin, MA, USA). Spectral-domain OCT EDI was acquired using the Spectralis instrument. Two graders independently analyzed OCT images to identify CCs. These graders were masked as to the patient’s diagnosis and the involved eye. If putative CCs were identified, then confirmation that they were CCs was performed by comparison to ICGA. CCs were defined as focal hyporeflective spaces with a characteristic posterior hypertransmission tail on individual B-scans, with a concordant lack of choroidal perfusion on ICGA. Disagreements were resolved through arbitration from a third reviewer; only two disagreements arose and had to be resolved. Cohen’s Kappa coefficient was 0.83. Representative imaging of CCs from a patient with multifocal choroiditis is shown in Figure 1, while Figure 2 shows CCs in patient with acute posterior multifocal placoid pigment epitheliopathy (APMPPE). Subfoveal choroidal thickness was measured in each eye using the caliper tool provided in the Spectralis software as previously described [8,15].

Multi-level mixed-effect logistic regression models were used to analyze associations between clinical and demographic parameters and the presence of choroidal caverns. In these models, the units of analysis were considered the eyes (at a lower level), which were nested within patients (at a higher level). Multi-level mixed effect regression was used specifically because it allows and appropriately accounts for inclusion of data from both eyes of a patient with bilateral disease [16]. Univariate analyses were initially performed for all parameters, and those with a *p*-value ≤ 0.250 [17] were included in the initial multivariate model. A backward elimination procedure was then performed to achieve the multivariate model presented. Anterior chamber and vitreous cell were dichotomized according to presence or absence of inflammation. All statistics were performed using Stata^®^ version 14.2 (StataCorp LP, College Station, TX, USA), and *p*-values < 0.05 were considered to be statistically significant.

## 3. Results

A total of 251 eyes from 135 patients were included in the study. Table 1 presents their clinical and demographic characteristics. The various uveitic diagnoses are listed in Table 2 with their associated subfoveal choroidal thickness.

Choroidal caverns were observed in 21 eyes (10.8%) (CC+ eyes) with posterior uveitis and 4 eyes (7.8%) with panuveitis. Caverns were observed only in patients who had bilateral uveitis. Among these patients with bilateral uveitis and CCs, 15 patients had CCs in one eye [15 CC+ eyes (60%)] only, while 5 patients had CCs in both eyes [10 CC+ eyes (40%)]. Caverns were not seen in any eyes with anterior or intermediate uveitis. Overall, the prevalence of CC+ eyes was 10%. We also examined the imaging of the unaffected eyes for the 19 patients with unilateral uveitis, and no CCs were observed in the unaffected eyes, as would be expected.

The distribution of various uveitis diagnoses in which CCs were found is shown in Table 3. Prevalence was calculated for the specific condition if it occurred frequently enough (>5 eyes with the condition in the study). Among the diagnoses for which there were at least five eyes included in the study, multifocal choroiditis (MFC) was the type of uveitis where CCs were most frequently observed, with 40% of eyes with MFC having CCs.

Clinical and imaging characteristics according to the presence and absence of CCs are listed in Table 4. In univariate analyses, only the male sex was significantly associated with the presence of CCs (*p* = 0.024), with men being more likely to have CCs than women. Vitreous cells were present at a similar rate in the two groups: 12 (48%) CC+ eyes vs. 111 (48.7%) CC− eyes. Five (20%) CC+ eyes were treated topically, 15 (60%) were treated systemically, and 5 (20%) were untreated. This contrasted to 36 (15.7%) CC– eyes that were treated topically, 87 (38.2%) systemically, and 104 (45.6%) untreated. In addition, the duration of disease was also similar between CC+ and CC− eyes (*p* = 0.077). For our multivariate analysis, three variables met our criteria for inclusion in the model: sex, systemic treatment, and duration of disease. Mean subfoveal choroidal thickness was also included. In the multivariate analysis, the male sex was still associated with CCs (OR 4.76, *p* = 0.012), while the duration of disease, systemic treatment, and choroidal thickness was not associated with CCs (*p* = 0.15, *p* = 0.06, *p* = 0.74, respectively).

## 4. Discussion

Several studies have estimated the presence of CCs in various retinal pathologies: in AMD, 12.5% of eyes exhibited CCs, while in pachychoroid disease, 52% of eyes had at least one cavern. This study presents, to our knowledge, the first assessment of CCs in eyes with uveitis. Our data identified CCs in 10% of eyes with non-infectious uveitis—notably, in 10.8% of eyes with posterior uveitis and 7.8% of eyes with panuveitis. CCs likely represent a degenerative process involving the choroid in eyes with AMD [8] or perturbations in choroidal vascular circulation in eyes with pachychoroid [10]. In this study, it is not surprising that CCs were found only in eyes with posterior and panuveitis, suggesting that direct choroidal/retinal inflammation is necessary to induce similar structural and/or vascular perturbations.

We found a relatively diverse population of uveitis that exhibited choroidal caverns; in the posterior uveitis subpopulation, MFC had the greatest percentage of eyes with CCs (as compared to all other uveitic diseases that were present in a minimum of 5 eyes). When compared with some other forms of posterior uveitis, MFC is associated with a higher degree of choroidal inflammation, as demonstrated by high rates of structural damage, including choroidal neovascularization, and often requires systemic immunosuppression [18,19,20]. It may be that the higher prevalence of CC+ eyes with MFC reflects a higher burden of inflammation leading to choroidal structural perturbations in this disease, although it is not possible to definitively conclude this given the limited number of eyes with each subtype of posterior uveitis. All patients with panuveitis and CCs had idiopathic uveitis (*n* = 4), and this number was so limited that subgroup analyses were not performed because of insufficient power.

The only identified parameter with a statistically significant association with the presence of CCs was the male sex. This association is of uncertain significance. In pachychoroid disease entities with CCs, ~91% of patients were male [8], while the single case report of CCs in Best vitelliform macular dystrophy was in a 15-year-old boy [11]. However, in patients with geographic atrophy from AMD, there was no sex predilection observed [10]. Some uveitides are more common in women, while others are more common in men [21]. Behcet’s disease and APMPPE have been shown to be more common in men in some studies [22]. The mechanisms for the gender differences in a predilection for uveitis are not well understood but may be due to hormone-mediated modulation [23], X chromosome inactivation [24], and various immunologic pathways [25]. These same gender-specific effects that influence disease occurrence may also influence pathogenic aspects of disease evolution, such as the formation of CCs. Further work is required to determine how/if sex is a risk factor for CCs in patients with uveitis.

Prevalence of CCs appeared higher in patients of Asian ethnicity as opposed to Caucasian, although regression analysis showed no statistical significance regarding CCs and ethnicity. Our study, however, had a limited number of Asian patients with NIU. A review of the literature does not show any predilection for CCs and ethnicity, although most published cohorts have been predominantly Caucasian.

The current study has several limitations, including the retrospective nature of its analysis. The heterogeneity of the patients, who were at various stages of their disease at the time of imaging, as well as the chronicity of treatment, may bias the prevalence estimates, although we adjusted for the duration of disease in the multivariate analyses. The estimates of CC prevalence may also be biased by the tertiary care nature of the population. This limits the generalizability of our findings to all forms of uveitis.

Posterior uveitis and panuveitis were over-represented due to the inclusion criteria that necessitated both OCT and ICGA, which would not have been routinely acquired in patients with only anterior or intermediate uveitis. Further, the number of anterior and intermediate uveitis patients was small, so it may be that CCs exist in these diseases. However, anterior and intermediate uveitis does not perturb the choroid to the same degree as posterior uveitis. Larger studies are required to further explore CCs in anterior and intermediate uveitis. Additionally, ICGA may be acquired in more severe forms of uveitis that could potentially bias the rate of CCs.

Given the cross-sectional nature of the study, further longitudinal work is needed to evaluate any potential evolution of caverns in eyes with uveitis. Intraocular inflammation was assessed as anterior chamber, vitreous cell, and choroidal thickening at the time of the initial imaging. Therefore, it does not capture other rarer markers of disease severity, such as optic nerve inflammation or the severity over the course of the disease. Ideally, eyes would be treatment naïve and subsequently followed with imaging to determine possible evolution of CCs pre- and post-treatment as well as CC association with intraocular inflammation and treatment routes. Future studies using Swept Source OCT should measure the size [26] and quantify the number of caverns, as well as determine if there are associations between size/quantity and various uveitides.

In summary, our study demonstrates that CCs are present in approximately 10% of eyes with posterior and panuveitis and are particularly common in eyes with MFC. Male sex may increase the risk for these lesions. Exploration of CCs in further pathologic and imaging studies may help with further understanding of the consequences of choroidal inflammation from different uveitides.

## Figures and Tables

**Figure 1 biomedicines-11-01268-f001:**
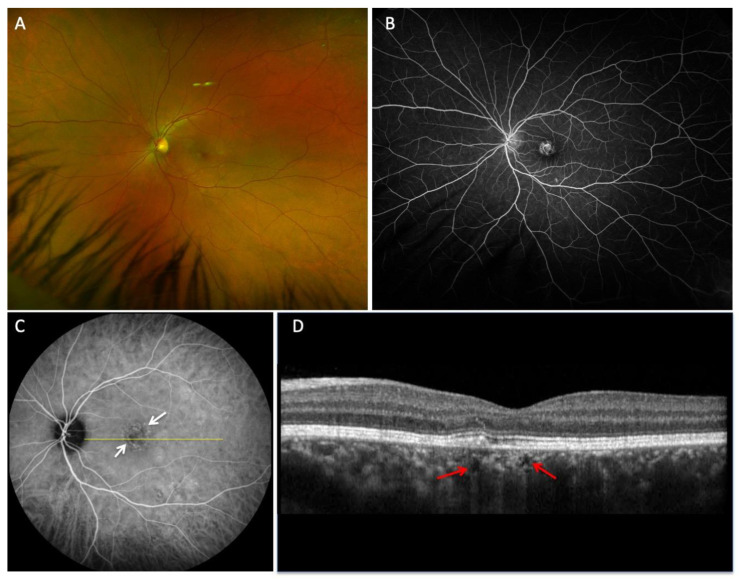
Representative images of a young woman with multifocal choroiditis and multiple choroidal caverns. (**A**), Wide-field fundus photos demonstrating subtle macular and mid-peripheral pigmentary changes in the left eye. (**B**), Fluorescein angiography showing late phase staining of the perifoveal region. (**C**), Late phase ICGA shows focal circular areas of hypofluorescence (white arrows), that localize with (**D**) to reveal several choroidal caverns. (**D**), Horizontal OCT B-scan, shows two choroidal caverns (red arrows) with characteristic posterior hypertransmission.

**Figure 2 biomedicines-11-01268-f002:**
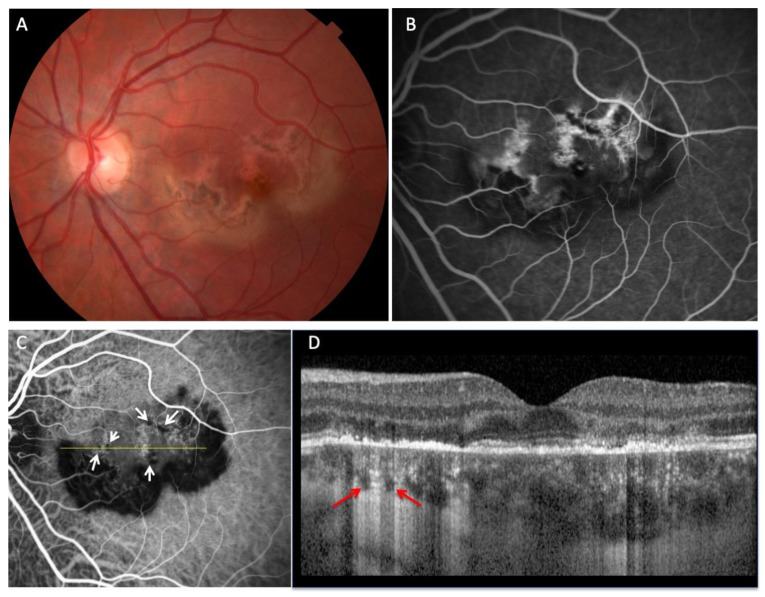
Multimodal imaging of a patient with posterior multifocal placoid pigment epitheliopathy (APMPPE) demonstrating multiple choroidal caverns. (**A**), Fundus photo demonstrating macular pigmentary changes surrounding areas of atrophy. (**B**), Fluorescein angiography demonstrates staining of the outer retinal lesions. (**C**), Late phase ICGA demonstrates multiple focal areas of hypofluorescence (white arrows), that co-localize with choroidal caverns ((**D**), horizontal OCT B-scan, red arrows).

**Table 1 biomedicines-11-01268-t001:** Demographic and clinical characteristics of patients with non-infectious uveitis.

Characteristic	Number or Mean
Number of eyes (patients)	251 (135)
Mean age, range (years)	48.8, 13–85
Female (patients)	103 (76.3%)
Race (patients)	
White	105 (77.8%)
African American	12 (8.9%)
Hispanic	4 (2.9%)
Asian	4 (2.9%)
Unknown	10 (7.5%)
Location of Uveitis (eyes)	
Anterior	1 (0.4%)
Intermediate	5 (2.0%)
Posterior	194 (77.3%)
Panuveitis	51 (20.3%)
Snellen Visual Acuity at Time of Imaging (eyes)	
20/15–20/40	183 (72.9%)
20/50–20/400	52 (20.7%)
<20/400	16 (6.4%)

**Table 2 biomedicines-11-01268-t002:** Non-infectious uveitis disease location and subtype.

Uveitic Diagnosis	Number of Eyes	Mean Subfoveal Choroidal Thickness um (SD)
Anterior Uveitis		
Idiopathic	1	288
Intermediate Uveitis		
Multiple Sclerosis	1	193
Idiopathic	4	251 (18.4)
Posterior Uveitis		
APMPPE	10	287.8 (36.4)
AZOOR	13	207.1 (68.2)
Behcet’s Disease	2	206 (18.4)
Birdshot Chorioretinopathy	42	213.2 (75.8)
Systemic Lupus Erythematosus	4	127 (2.8)
MEWDS	5	278.4 (116)
MFC	15	263.8 (101.7)
PIC	2	189 (22.6)
POHS	4	221.7 (22.9)
IRVAN	2	-
Sarcoidosis	8	200.8 (59.3)
Serpiginous Choroiditis	2	-
GPA	2	162 (2.8)
Idiopathic	83	274.2 (100.3)
Panuveitis		
Multiple Sclerosis	2	252.5 (24.8)
VKH	4	-
Sarcoidosis	4	212 (45.3)
Idiopathic	41	217.8 (98)

APMPPE, acute posterior multifocal placoid pigment epitheliopathy; AZOOR, acute zonal occult outer retinopathy; MEWDS, multiple evanescent white dot syndrome; MFC, idiopathic multifocal choroiditis; PIC, punctate inner choroiditis; POHS, presumed ocular histoplasmosis syndrome; IRVAN, idiopathic retinal vasculitis, aneurysms, and neuroretinitis; GPA, granulomatosis with polyangiitis; VKH, Vogt-Koyanagi-Harada syndrome. SD, standard deviation.

**Table 3 biomedicines-11-01268-t003:** Non-infectious uveitis diagnoses in eyes with choroidal caverns.

Diagnosis	Number of Eyes	CC Prevalence within Uveitis Subtype
Posterior Uveitis		
APMPPE	1	10%
Behcet’s Disease	1	*
Birdshot Chorioretinopathy	2	4.8%
Systemic Lupus Erythematosus	2	*
MEWDS	1	20%
MFC	6	40%
Serpiginous Choroiditis	1	*
GPA	1	*
Idiopathic	6	7.8%
Panuveitis		
Idiopathic	4	9.8%

* Not calculated, less than 5 eyes with diagnosis included in study. APMPPE, acute posterior multifocal placoid pigment epitheliopathy; MEWDS, multiple evanescent white dot syndrome; MFC, idiopathic multifocal choroiditis; GPA, granulomatosis with polyangiitis.

**Table 4 biomedicines-11-01268-t004:** Univariate analysis of demographic and clinical parameters in eyes with presence vs. absence of choroidal caverns.

	Eyes + for Choroidal Caverns	Eyes − for Choroidal Caverns	Odds Ratio/Beta Coefficient ^#^	*p*-Value
Mean age in years (SD)	46.1 (16.2)	49.2 (17.3)	−0.02	0.461
Sex (% male)	12 (48.0%)	48 (21.2%)	6.67	**0.024**
^a^ Race				
White	18 (72%)	177 (78.3%)		
Asian	3 (12%)	4 (1.8%)	21.64	0.070
Black	4 (16%)	18 (8.0%)	4.08	0.198
Hispanic	0	7 (3.1%)	*	*
Unknown	0	20 (8.8%)	*	*
Mean duration of disease in months (SD)	54.5 (83.1)	28.7 (48)	0.01	0.077
Presence of anterior chamber cell	2 (8%)	47 (20.6%)	0.26	0.251
Presence of vitreous cell	12 (48%)	111 (48.7%)	1.11	0.885
^b^ Local treatment	5 (20%)	36 (15.9%)	2.87	0.325
^b^ Systemic treatment	15 (60%)	85 (37.6%)	3.59	0.094
Mean subfoveal choroidal thickness in μm (SD)	226.5 (90.8)	242.0 (90.8)	−0.002	0.526

SD, standard deviation.^#^ Beta coefficient is reported for continuous variables, while odds ratio is provided for dichotomous or categorical values. ^a^ Reference group is White. ^b^ Reference group is no treatment. * Unable to compute due to lack of observations in CC+ eyes cohort.

## Data Availability

The data presented in this study are available on request from the corresponding author. The data are not publicly available due to possible patient identifiers.

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
