# Peer review of "Presence of Choroidal Caverns in Patients with Posterior and Panuveitis"

_biomedicines, 2023, doi:10.3390/biomedicines11051268_

Round 1

Reviewer 1 Report (New Reviewer)

The manuscript presented for evaluation is interesting and indicates a potentially new symptom visible in OCT scans, concerning mainly MFC. More and more papers describe CC in various diseases of the retina and choroid, not only those mentioned by the Authors.

Why were several patients with inflammation in the anterior or intermediate segment of the eye included in the study? They are not even worth mentioning - it may give the impression that there are no CCs in anterior and intermediate inflammation. In fact, the manuscript does not provide any reliable information about it.

Wasn't choroidal thickness more important than type of inflammation (e.g. MFC)? What was the mean choroidal thickness in each disease? The choroidal thickness is given only in the fovea, perhaps the thickness in the vicinity of the CC is more important. I know that probably multi-level mixed effect logistic regression models were also supposed to exclude this type of relationship. But I'm not at all convinced that they did.

Once the Authors answer the above questions or include these comments in the limitations, the manuscript may be published.

Author Response

Reviewer 2 Report (New Reviewer)

This manuscript by Begaj et al found that some patients with chronic non-infectious uveitis (NIU) had choroidal caverns (CC) for the first time using optical coherence tomography and indocyanine green angiography. The data are solid and clear.  It is suitable for publication with minor revision. I have some minor comments that need to be addressed first.

1.     The authors used 4 Asian patients while 105 patients are white (from table 1). From table 4, it showed the presence of CC in 18 eyes of 105 white patients, while the presence of CC in 3 eyes of 4 Asian patients, I understand the authors cannot draw any conclusion on whether there is any racial difference in CC prevalence especially in Asian patients with NIU due to limited number of Asian patients. But if there is any racial difference in the prevalence of CC in other eye diseases, please give a brief discussion.

2.     It is well established that many physiological functions of eye are accompanied with circadian changes, including the choroidal thickness. I was wondering whether CC many also showing some circadian changes?

Thank you!

Author Response

Reviewer 3 Report (New Reviewer)

I consider this study to have valuable data that would be of interest if published. The study showed for the first time that chroidal caverns (CC) are present in approximately 10% of eyes with posterior and panuveitis and are particularly common in eyes with multifocal choroiditis. Male sex may increase the risk for these lesions. Exploration of CC in further pathologic and imaging studies may help with further understanding of the consequences of choroidal inflammation from different uveitides. 

Author Response

We thank the reviewer for your kind comments and appreciation of our work.

This manuscript is a resubmission of an earlier submission. The following is a list of the peer review reports and author responses from that submission.

Round 1

Reviewer 1 Report

Recently, the studies about CC have been increased.

As you mentioned in your article, this is the first study to describe CC in uveitis.

But the important thing is that this research design is inappropriate. 

So it is unreasonable to apply this result to all uveitis. 

And there are some parts which is revised.

1. (J Clin Med  2022 Nov 26;11(23):6994.) Xiaohong etc presented the classification of CC. Did you compare the type of CC of uveitis ? 

2.  There is no sub-group analysis  of Panuveitis in table 3. 

You should describe the subgroup analysis in table. 

3. Did you compare pretreatment status with posttreatment status in CC?

4. In your study, you described in discussion 

" MFC had the greatest percentage of eyes with CC . When  compared with some other forms of posterior uveitis, MFC is associated with a higher  degree of choroidal inflammation as demonstrated by high rates of structural damage including choroidal neovascularization and often requires systemic immunosuppression "

However this may vary depending on the activity or severity of uveitis.

Did you compare the severity of the same condition and same disease?

    •  

Author Response

Review 1:

Recently, the studies about CC have been increased.
As you mentioned in your article, this is the first study to describe CC in uveitis.
But the important thing is that this research design is inappropriate. 
So it is unreasonable to apply this result to all uveitis. 

We appreciate the reviewer’s comments. We have only examined a single tertiary care institution’s uveitis population so that is certainly a limitation and thus not generalizable to all uveitis. We have added this limitation to the Discussion (lines 194-5)

And there are some parts which is revised.

  1. (J Clin Med 2022 Nov 26;11(23):6994.) Xiaohong etc presented the classification of CC. Did you compare the type of CC of uveitis? 

Thank you for pointing out this excellent paper. It was published after our data collection.  It used Swept-Source OCT on the Svision platform for identification of the caverns, while our study was done with Spectral Domain OCT on Heidelberg Spectralis Platform.  We have not quantified the size of caverns, and may have less resolution with Spectral Domain OCT to detect all of them when compared to Swept Source OCT, particularly for smaller caverns. We have added the paper as a reference and included a future direction of research to investigate the number of caverns using Swept Source OCT (lines 203-205).

  1. There is no sub-group analysis  of Panuveitis in table 3.
    You should describe the subgroup analysis in table. 

There were only four patients with panuveitis and choroidal caverns and they all had idiopathic panuveitis.  We have previously attempted subgroup analyses but the numbers are too low to make any meaningful conclusions. We have included this limitation in lines 177-179.

  1. Did you compare pretreatment status with posttreatment status in CC?

We did not have sufficient numbers of patients with the necessary imaging to compare pretreatment to posttreatment CC status. We did have enough patients to compare patients that were on systemic treatment vs. those without treatment with our cross-sectional case-control design. We agree that future studies should look at longitudinal data, pre- and post-treatment, to see how treatment affects caverns. We have specified this limitation further in the Discussion (line 204).

  1. In your study, you described in discussion 

" MFC had the greatest percentage of eyes with CC . When compared with some other forms of posterior uveitis, MFC is associated with a higher  degree of choroidal inflammation as demonstrated by high rates of structural damage including choroidal neovascularization and often requires systemic immunosuppression "

However this may vary depending on the activity or severity of uveitis.
Did you compare the severity of the same condition and same disease?

We did examine intraocular inflammation (anterior chamber and vitreous cell, as indicated by the anatomic location of uveitis) and choroidal thickening as markers of severity for association with choroidal caverns and did not find an association (Table 4 and lines 141-142).  We acknowledge that there is significant clinical heterogeneity, with varying sites of involvement in uveitis and also some other less common markers of disease severity (e.g. optic nerve inflammation, outer retinal attenuation, etc).  So it is difficult to precisely quantify disease severity between patients, but we tried to do so with the most common markers of disease severity.  We have added these points to the Discussion (lines 199-201).

Reviewer 2 Report

The study suffers from definition of Choroidal Caverns, and also the readers should be blinded to controls and to uveitis cases. The figures show no definite caverns which typically are huge spaces, while here the two figures show mini-cavern or is it  cross section through a feeding vessel? I am not at ease to define the findings noted as caverns. The size is very important

Author Response

Reviewer 2

The study suffers from definition of Choroidal Caverns, and also the readers should be blinded to controls and to uveitis cases. The figures show no definite caverns which typically are huge spaces, while here the two figures show mini-cavern or is it  cross section through a feeding vessel? I am not at ease to define the findings noted as caverns. The size is very important

We are sorry we about this omission – the two readers were masked to the patient’s underlying condition when they reviewed the OCTs.  They did not know the patient’s diagnosis, including type of disease or involved eye.  We have clarified this in the Methods (lines 80-82).

A recent publication (https://pubmed.ncbi.nlm.nih.gov/36498569/) looked at caverns in various conditions (age-related macular degeneration, central serous chorioretinopathy, etc). The majority of caverns were small (a figure from the paper is shown below) and of similar size to the choroidal caverns we saw, which are in Figure 1 and 2 of our paper. To ensure that the lesions we identified on OCTA were truly CC and not other pathologies, such as feeding vessels, we necessitated the use of ICGA to ensure that the space seen on OCT was indeed a cavern by colocalizing an area of choroidal nonperfusion to the cavern.  Furthermore, we confirmed a posterior hypertransmission tail on the OCT (lines 82-4).

Figure 1 from Guo X, Zhou Y, Gu C, Wu Y, Liu H, Chang Q, Lei B, Wang M. Characteristics and Classification of Choroidal Caverns in Patients with Various Retinal and Chorioretinal Diseases. J Clin Med. 2022 Nov 26;11(23):6994. doi: 10.3390/jcm11236994. PMID: 36498569; PMCID: PMC9740557. Yellow arrow is a choroidal vessel while red and white are choroidal caverns that are similar in size to those in our study.

Round 2

Reviewer 1 Report

This research design is inappropriate. 

And there are many flaws in this article.

  •  

Additional Comments:

There are many flaws in this article . 1. This research design is inappropriate. 2. Additional experiment should be added. 3. When seen the your response on first review, you have repeatedly replied that the small number of subjects in your study is problematic. So It seems that the problems of this paper could not be solved.  

Reviewer 2 Report

The authors have analysed further their novel findings and it seems that the caverns are more common in longer duration uveitis and more so in Asians. This reminds me of polypoidal choroidal vasculopathy being more prevalent in Asians and the future would tell us if th two entities are related

Author Response

We appreciate the reviewer's word and thank them for them time in order to help our manuscript.

The polypoidal link is very interesting and something worth studying in the future!